# "Sickness has no time": Awareness and perceptions of health care workers on universal health coverage in Uganda

Susan C. Ifeagwu[1]*, Ruth Nakaboga Kikonyogo[2], Suzan Nakkazi[3], Joshua Beinomugisha[3], Stephen Ojiambo Wandera[4], Suzanne N. Kiwanuka[5], Rachel King[6], Tine Van Bortel[7,8], Carol Brayne[7], Rosalind Parkes-Ratanshi[1,3]

1 Cambridge Public Health, Department of Public Health and Primary Care, University of Cambridge, Cambridge, United Kingdom, 2 Infectious Diseases Institute, Makerere University College of Health Sciences, Kampala, Uganda, 3 The Academy for Health Innovation, Infectious Diseases Institute, Makerere University College of Health Sciences, Kampala, Uganda, 4 Department of Population Studies, School of Statistics and Planning, College of Business and Management Sciences, Makerere University, Kampala, Uganda, 5 Health Policy Planning and Management, Makerere University School of Public Health, Kampala, Uganda, 6 Epidemiology and Biostatistics, University of California, San Francisco, San Francisco, California, United States of America, 7 Cambridge Public Health, Department of Psychiatry, University of Cambridge, Cambridge, United Kingdom, 8 Leicester School of Applied Health Sciences, Faculty of Health and Life Sciences, De Montfort University, Leicester, United Kingdom

* sci24@cam.ac.uk

**Data Availability Statement:** All relevant data are within the manuscript and its Supporting Information files.

## Abstract

### Introduction

Each person having access to needed health services, of sufficient quality, and without suffering financial hardship, defined as universal health coverage (UHC) by the World Health Organization, is critical to improve population health, particularly for vulnerable populations. UHC requires multisectoral collaboration and good governance, and this will require buy-in of key stakeholders; but their views are under-documented. The aim of this stakeholder analysis was to explore the awareness and perceptions of UHC by health care workers (HCWs) in Uganda.

### Methods

A mixed-methods study was conducted based on primary data from HCWs including an online Qualtrics[XM] survey of 274 HCWs (from a database of persons who had received training at an academic institution), 23 key informant semi-structured interviews, and one eight-person focus group discussion. Data was collected from February to April 2022. Microsoft Excel and R Programme were used for quantitative analyses and NVivo version 12 for qualitative analyses.

### Results

HCWs attributed a high level of importance to UHC in Uganda. Participants discussed national communication and management practices, organisational roles, health financing and power dynamics, health care demand and the impact of and learnings from COVID-19.

**Funding:** The author(s) received no specific funding for this work.

**Competing interests:** All authors declare no competing interests.

**Abbreviations:** AMR, Antimicrobial Resistance; COVID-19, Coronavirus disease 2019; FGD, Focus Group Discussion; GDP, Gross Domestic Product; HC, Health Centre; HCW, Health Care Worker; IDI, Infectious Diseases Institute; M&E, Monitoring and Evaluation; NCD, Non-Communicable Disease; NGO, Non-Governmental Organisation; NMS, National Medical Store; OOP, Out-of-pocket; PHC, Primary Health Care; PNFP, Private-Not-For-Profit; PPP, Public-Private Partnership; REC, Research Ethics Committee; SDGs, Sustainable Development Goals; SSA, Sub-Saharan Africa; UHC, Universal Health Coverage; UN, United Nations; WHO, World Health Organization; XM, Experience Management.

Four main themes–each with related sub-themes–emerged from the interview data providing insights into: (1) communication, (2) organisation, (3) power, and (4) trust.

## Conclusion

There is a critical need for better communication of UHC targets by policymakers to improve understanding at a grassroots level. Results indicated that ensuring trust among the population through transparency in metrics and budgets, strong accountability measures, awareness of local cultural sensitivities, sensitisation of the UHC concept and community inclusion will be essential for a multisectoral roll out of UHC. Further provision of quality health services, a harmonisation of efforts, increased domestic health financing and investment of HCWs through fair remuneration will need to underpin the delivery of UHC.

## Introduction

Universal health coverage (UHC) is defined by the World Health Organization (WHO) as all individuals and communities having access to any health services they need, of sufficient quality to be effective, without suffering financial hardship [1]. Uganda, a low-income country in East Africa [2], remains committed to achieving UHC, as stipulated in the ministerial Roadmap for UHC. However, despite well delineated national strategic objectives, gaps remain in the perceptions and awareness of key stakeholders in the country [3].

UHC relies on strong collaboration, effective governance and leadership, with good understanding and cooperation among actors or stakeholders in a health system [4, 5]. The multisectoral nature of UHC and breadth of stakeholders involved add complexity to the implementation of strategies for UHC [6, 7]. As a research tool, stakeholder analyses can help implementers and policymakers gain insights on a relevant topic or policy from key actors or stakeholders in health reforms [8]. Gilson and colleagues provided a basis for conducting stakeholder analyses to study UHC or "universal coverage" reforms, by sharing lessons learnt on how such analyses have been used to support the development of pro-poor health policies and potential strategies for political management in South Africa and Tanzania. Stakeholder analyses can assist in understanding the political viability of policy proposals and help support management strategies concerning these policies. Therefore, authors proposed repeated stakeholder analyses to address the dynamics of policy change [8]. Given that stakeholders have deep local and practical knowledge, their involvement in strategic planning can produce higher quality, enhanced understanding, and more feasible policy decisions [9].

Within the literature, stakeholder analyses have been conducted in Uganda to assess the impact of user fee abolition, evaluate the implementation of policies related to communicable and non-communicable diseases (NCDs), and to inform health system strengthening [10–12]. Nonetheless, these analyses have not focused on perceptions and awareness of key stakeholders, particularly health care workers (HCWs), related to UHC in Uganda.

HCWs are the foundation of a well-functioning health system and quality health services, thus their direct involvement in health policy decisions is important. In the past, only several studies have looked at their perceptions and understanding related to UHC [13–15]. In 2016, Koon et al. [14] conducted semi-structured interviews with 60 nurses based in three health facilities in Kenya. Findings revealed that nurses had not heard of UHC and were unfamiliar

with the concept. The authors argued that nurses should be involved as stakeholders for successful adoption of UHC initiatives.

Whilst studies related to HCWs' perceptions of UHC have not yet been conducted in Uganda, Essue and Kapiriri (2021) evaluated priority setting for health system strengthening in Uganda through a policy document review and interviews of stakeholders [12]. Identified challenges concerning the implementation of priority programmes included: parallel priority setting processes by development assistance partners that were not aligned with the Ministry of Health, earmarked funding, and a lack of domestic health financing. The authors emphasised the need to better coordinate resources across silos and allow for improved health system efficiency.

Presently, there is limited information on HCW perceptions of UHC and how their personal values affect their attitudes towards UHC in Uganda. Due to the multisectoral nature of UHC, wide stakeholder engagement will be essential to achieve UHC. Through investigating these perceptions, we hypothesized that further knowledge for implementation practices could be gleaned. This paper presents the results of the perceptions and awareness of HCWs on UHC in Uganda explored through a mixed-methods study.

## Methods

### Study design

For this study, the mixed-methods design included an online survey, semi-structured interviews, and a focus group discussion (FGD). The study population of the online survey included local, national, regional, and international stakeholders who worked in UHC, health policy, financing, and systems research. For the semi-structured interviews and FGD, the study population included Ugandan health care workers. Guiding this study, the underlying conceptual framework was informed by Walt and Gilson's (1994) policy triangle framework, which described the complex interplay between context, content, process, and actors in shaping health policy and influencing policy implementation [16]. An original user-friendly, online email survey was developed by the researchers, to examine stakeholders' perceptions of UHC and enhance survey accessibility [15, 17–19]. This was guided by the implementation strategy stipulated by the WHO Consultative Group on Equity and UHC [20]. No relevant prior instrument was found; thus, the online survey was developed by the researchers specifically to answer the specific research questions. The online email survey explored perceived challenges to reaching UHC, perceptions of and priority areas for UHC, awareness, values, strategies, and best practises among international, regional, and national stakeholders.

The survey was created on the online software Qualtrics experience management (XM) platform (Qualtrics International Incorporated, Seattle, Washington, USA). A pilot test of the online survey was conducted between 1 and 5 February 2021, which included a survey evaluation form shared with eight staff members at the Infectious Diseases Institute (IDI), a non-governmental organisation owned by Makerere University in Uganda. Study questions were further refined based on feedback from the pilot group to provide more clarity and correct any technical issues. The final online survey comprised an introductory section, including background information, a consent form, and 26 questions (S1 Appendix). The survey included closed-ended and open-ended questions, such as single choice items, binary questions ('yes/no' options), drop-down menus, multiple choice items, Likert scale questions, free text entry fields, and a scored question. Randomisation was used for multiple-choice questions to avoid potential order or selection bias.

For the online email survey to all stakeholders, which was based on the entire research project, the following sample size calculation was incorporated. The calculation was based on the

$$\text{Sample size } (n) = \frac{N}{1 + N(e^2)} \times \frac{1}{r}$$

$$n = \frac{1000}{1 + 1000(0.05^2)} \times \frac{1}{0.95}$$

$$n = \frac{1000}{3.5} \times 1.05$$

$$n = 300.7518 \dots$$

$$n = 301$$

| | |
|---|---|
| $n =$ | Desired sample size |
| $N =$ | Total population of participants in distribution list/UHC work |
| $e =$ | precision (0.05) |
| $r =$ | response rate at 95% |

**Fig 1. Sample size calculation for the online UHC survey.**

primary objective of the importance of UHC, which was measured by a 5-point Likert scale question [21, 22]. Given the estimated population size of 1,000, which is an approximation and best estimate based on the distribution list and purposive sampling, the Yamane's sample size determination formula (1967) was applied and adapted by the consideration of a response rate of 95% (Fig 1) [23].

In this formula, where a 95% confidence level and $p = 0.5$ were assumed, '$n$' is the sample size, '$N$' is the population size and '$e$' is the level of precision at 0.05 [24]. Therefore, applying this formula to the parameters of the project produced a sample size of $n = 301$ participants. This sample size aimed to address questions related to the perceived importance of UHC, for which response options included "very important", "quite important", "fairly important", "not important", or "not applicable".

HCWs were selected on the basis that they had previously attended training courses conducted at IDI and had expressed willingness to be contacted for further feedback. The survey was circulated to various Government of Uganda institutes, Private Not-For-Profit (PNFP) institutions and private HCWs of different cadres including doctors, nurses, medical officers, laboratory technicians, administration, and management from the entire sub-Saharan African (SSA) region. The survey was disseminated in batches of 500 individuals, randomly selected from the IDI database, up until a sample size of at least 301 participants was reached, amounting to approximately 14,000 e-mail contacts. Individual emails were captured and tracked through the use of an external Excel list of contacts, which listed all email addresses and guaranteed that there were no duplications. The survey was initiated on 1 February 2022 and remained open for three months, up to 30 April 2022, and study participants were recruited throughout this period. Further potential participants were identified through the snowballing procedure. There was a reimbursement for internet data for individuals in Uganda who had consented to follow-up interviews of 10,000 Ugandan Shillings (approx. 2.5 USD) through mobile money transfer.

A final question of the survey asked whether respondents would be willing to be contacted for further participation in form of an interview. Individuals who registered their interest, either by providing their email address or telephone number, were added to the potential interview group, as part of the cohort for the qualitative methodological approach [18, 25]. Participants who indicated their interest in the in-depth semi-structured key informant interviews were selected through purposive sampling. Purposive sampling was used to begin with, during the online survey stage, and snowball sampling was conducted subsequently throughout the data collection phase and through discussions with experts, research team members and participants who were able to suggest further experts who were relevant participants considering our research questions. Additional interviewees were identified through the

snowballing technique to expand the breadth of informants included. Both methods were used to expand the reach of relevant participants and to save time, which were considered as strengths of using both approaches together. However, weaknesses exist with this pragmatic approach, which are discussed in the limitations section. The semi-structured interviews aimed for 20 to 30 participants, while the FGD of HCWs planned for eight to ten individuals. Individuals who provided their telephone numbers were called and requested to share their email address. Participant information sheets and consent forms were then sent to all participants for their deliberation and approval. Results from the online survey were analysed quantitatively and helped shape the composition of interview questions for the next stage.

Interviews of key informants were conducted between mid-February to the end of April 2022. All interviews were conducted online using Microsoft Teams version 4.11.12.0 (Microsoft Corporation, Washington, USA) or Zoom version 5.9.6 (4993) by Zoom Video Communications Incorporated, San Jose, California, USA. On average, interviews lasted 50 minutes, ranging from 30 to 65 minutes in total. Questions were based around the priorities of said stakeholders, their roles, understanding, barriers and challenges related to UHC, and next steps needed for a multisectoral approach tailored to the country context. The individual qualitative semi-structured key informant interviews were conducted until saturation or information power was achieved [26]. The notion of saturation, defined by Glaser and Strauss (1967) in their seminal work on grounded theory, rested on the concept that new observations were added to the main analysis and once these became repetitive (i.e. no new themes emerging), saturation was reached [27].

The FGD of eight Ugandan HCWs was conducted in person at IDI in Kampala on 7 March 2022 and lasted for about one hour. Participants obtained from the online survey and through snowballing, who were willing to attend in person, were recruited. Questions for the discussion delved into related online email survey responses and drew on topical areas similar to the semi-structured interviews. On the day of the FGD, which was largely obtained through the snowballing procedure, one discussant was solely able to speak the local language, Luganda. For the purpose of the group and to make the most of various viewpoints, it was decided to include the participant and translate any questions, probes, and responses directly to her to encourage participation. The FGD was facilitated by a moderator (SI) and an observer (SN), who translated answers provided in the local language, Luganda, to the moderator. Any nonverbal cues throughout the interview were also noted.

Upon completion of the interviews and FGD, verbatim transcriptions were developed based on audio recordings. All survey and interview data were analysed using Microsoft Excel version 16.54 (21101001) by Microsoft Corporation in Washington, USA, and R programming version 3.6.1 and RStudio version 1.2.1335 (R Core Team, University of Auckland, New Zealand).

## Ethics approval

Ethical approval was granted by the IDI Scientific Review Committee (Reference 31/2020), Makerere University School of Public Health REC (Reference: SPH-2021-33), Ugandan National Council for Science and Technology (Reference: HS1478ES), and Cambridge Psychology REC (Reference: PRE.2021.104). A data sharing agreement was developed and co-signed by the University of Cambridge and IDI on 18 February 2021 to enable sharing of data between organisations. All study participants received information sheets and consent forms, and formal consent was obtained through written informed consent. All recorded data and interview transcripts were anonymised through a generalised code to remove any personal

identifiers and ensure confidentiality both during and after data collection. Thus, authors had no access to identifiable information.

## Data analysis

**Statistical analysis.** Descriptive statistics and frequency tabulations were made for all survey responses and chi-square tests were used to assess any differences in urban and rural settings, and knowledge of UHC versus no knowledge of UHC.

**Semi-structured key informant interviews and FGD.** The qualitative data collected was analysed with guidance from Ritchie and Spencer's (1994) Framework Analysis, comprised of five stages, namely (1) familiarisation, (2) identifying a thematic framework, (3) indexing, (4) charting, and (5) mapping and interpretation [28]. The framework was guided by the WHO six building blocks of health systems framework on financing, health workforce, information, leadership and governance, medical products, and service delivery [4]. Data was reviewed repeatedly and categorised into codes using an inductive approach, and interpreted and organised into overarching themes [29, 30]. Stages one to four were carried out using NVivo version 12 (QSR International, Massachusetts, USA) and the use of an audit trail in form of memo writing, following similar methods to those employed by other software users [31]. Both the coding and theme development process were carried out by one researcher.

Coding involved a highly iterative and dynamic process, with initial open coding of the entire body of data reviewed and concepts revised and refined to establish the most applicable categories and codes [32]. Throughout the line-by-line coding process, multiple codes given to one sentence were flagged to help identify potential cross-cutting themes. Overlapping coding was reviewed for eligibility and similar concepts were merged into one code. Related codes were then grouped into themes. The final stage five was conducted using the coding framework matrix feature on NVivo 12 to enable a cross-tabulated matrix by coded themes and develop data interpretations. Findings were summarised and presented through a thematic synthesis, enabling the identification of gaps.

## Results

### Online survey

**Characteristics of HCW respondents.** In total, $n = 274$ HCWs completed the survey. More than half (56%) of all HCWs ($n = 154$), indicated that they worked in the government sector, followed by the private sector, academia, and non-governmental organisations (NGOs) (Table 1). Most respondents were based in an urban setting. Both urban and rural settings were derived from the health facility location. The most frequently reported important motivation in the day-to-day job of HCWs relative to other predetermined motivational factors was to serve their community (Fig 2).

**Perceptions and knowledge of UHC by HCWs.** Almost two thirds (66%) of HCWs indicated that they had knowledge of UHC, while approximately one third (34%) did not (S1 Table). Overall, 233 respondents (85%) selected the correct definition of UHC, while 41 selected a partial or incorrect definition. The internet was the main source of how respondents had first heard about UHC, followed by information from seminars, meetings, or conferences and from colleagues. Most respondents specified that they did not know anyone who worked in the area of UHC.

More than half of the respondents were aware of strategies related to UHC from the government or the Ministry of Health. Respondents who had knowledge of UHC reported awareness of strategies more frequently than those without UHC knowledge (69% compared to 30%, $p<0.001$, S2 Table). Respondents who had knowledge of UHC were more likely to report

**Table 1. Main characteristics of health care worker survey participants (n = 274).**

| Category | Count (as a %) |
|---|---:|
| Institution (n = 274) | |
| Government | 154 (56.2) |
| Private | 74 (27.0) |
| Academia | 11 (4.0) |
| Other* | 35 (12.8) |
| Total | 274 (100.0) |
| Region (Ugandan nationals only, n = 270) | |
| Central | 98 (36.3) |
| Eastern | 26 (9.6) |
| Northern | 79 (29.3) |
| Western | 67 (24.8) |
| Total | 270 (100.0) |
| Setting (n = 274) | |
| Rural | 80 (29.2) |
| Urban | 194 (70.8) |
| Total | 274 (100.0) |
| Facility type (for all countries, n = 274) | |
| Health Centre (HC) III, II or I | 80 (29.2) |
| HCIV | 45 (16.4) |
| Headquarters/Government Office (Non-clinical) | 14 (5.1) |
| National Hospital | 21 (7.7) |
| Private Not-For-Profit (PNFP) or Private Hospital/Clinic | 39 (14.2) |
| Regional Referral Hospital | 34 (12.4) |
| Other** | 41 (15.0) |
| Total | 274 (100.0) |

Note

*Categories for 'other' included: Non-Governmental Organisation (NGO) = 21 (7.7%); International Organisation = 3 (1.1%); Implementing Partner = 3 (1.1%); Private Not-For-Profit (PNFP) = 2 (0.7%); Private, academia and government = 1 (0.4%); Retired = 1 (0.4%); and unspecified = 4 (1.5%).

**Categories for 'other' included: Community-based organization = 1 (0.4%); District General Government Hospital = 10 (3.6%); Drop-in Centre = 1 (0.4%); Cross cutting (e.g. Health Centres, National Hospital, PNFPs) = 4 (1.5%); Laboratory and research clinic or institute = 4 (1.5%); National and PNFP Hospital, University = 1 (0.4%); NGO = 9 (3.3%); Private medical centre = 2 (0.7%); United Nations agency = 1 (0.4%); University = 3 (1.1%); and unspecified = 5 (1.8%).

awareness of national targets or goals for UHC than those without knowledge (50% and 23%, respectively, $p < 0.001$, S2 Table).

There was high agreement of over 90% across six priorities in terms of the most essential health priority for Uganda, with respondents 'somewhat' to 'strongly' agreeing with these factors (Fig 3). The six priorities included: (1) quality essential health service access, (2) addressing determinants of health (e.g. social, economic, environmental factors), (3) essential medicines, diagnostics, and devices availability for primary health care (PHC), (4) communicable diseases, (5) NCDs and (6) health promotion and information. Most respondents strongly agreed with access to quality essential health services and the availability of essential medicines, diagnostics, and devices for PHC as being the most essential health priority for Uganda. A reduction in the number of people suffering from financial hardships was

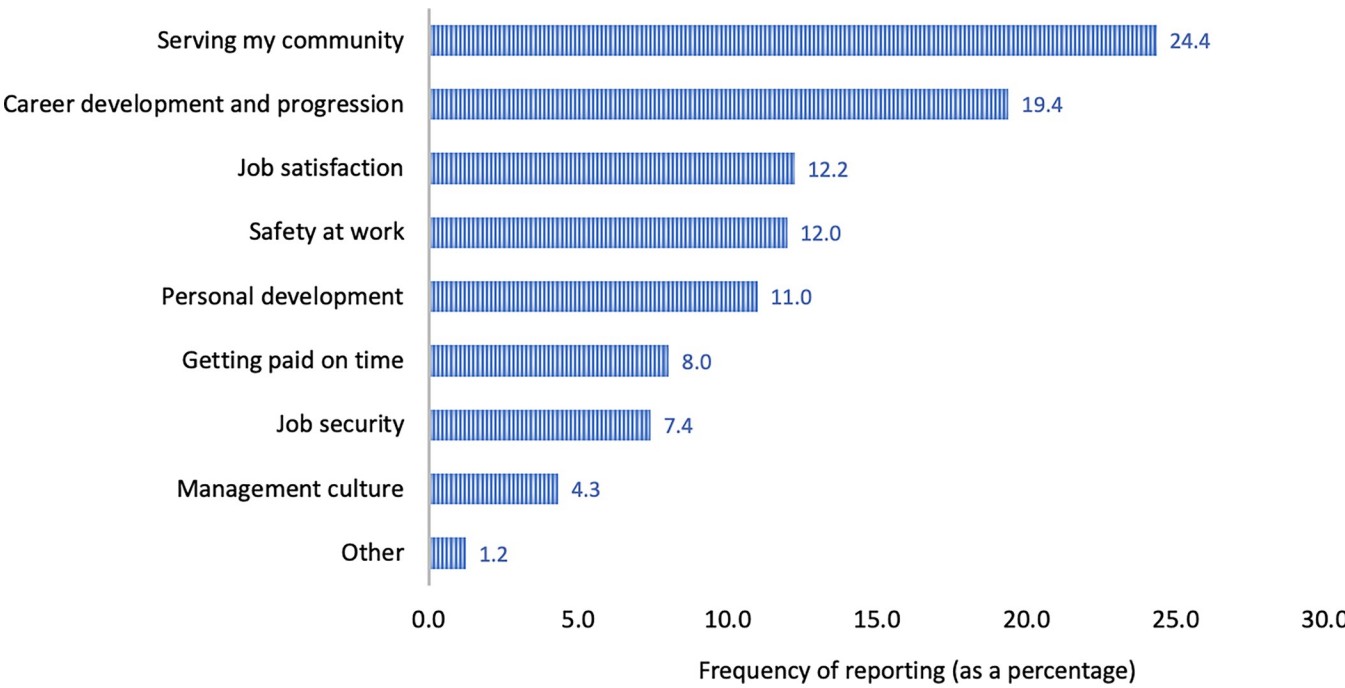

**Fig 2. Most important consideration in the day-to-day job of HCW participants (*n* = 274).**

considered the lowest priority by respondents, compared to the other factors, and the greatest disagreement was reported for this selection.

Overall, the perceived level of importance of UHC was high among HCW respondents. Almost all HCWs valued UHC as being either 'extremely', 'very', or 'moderately' important.

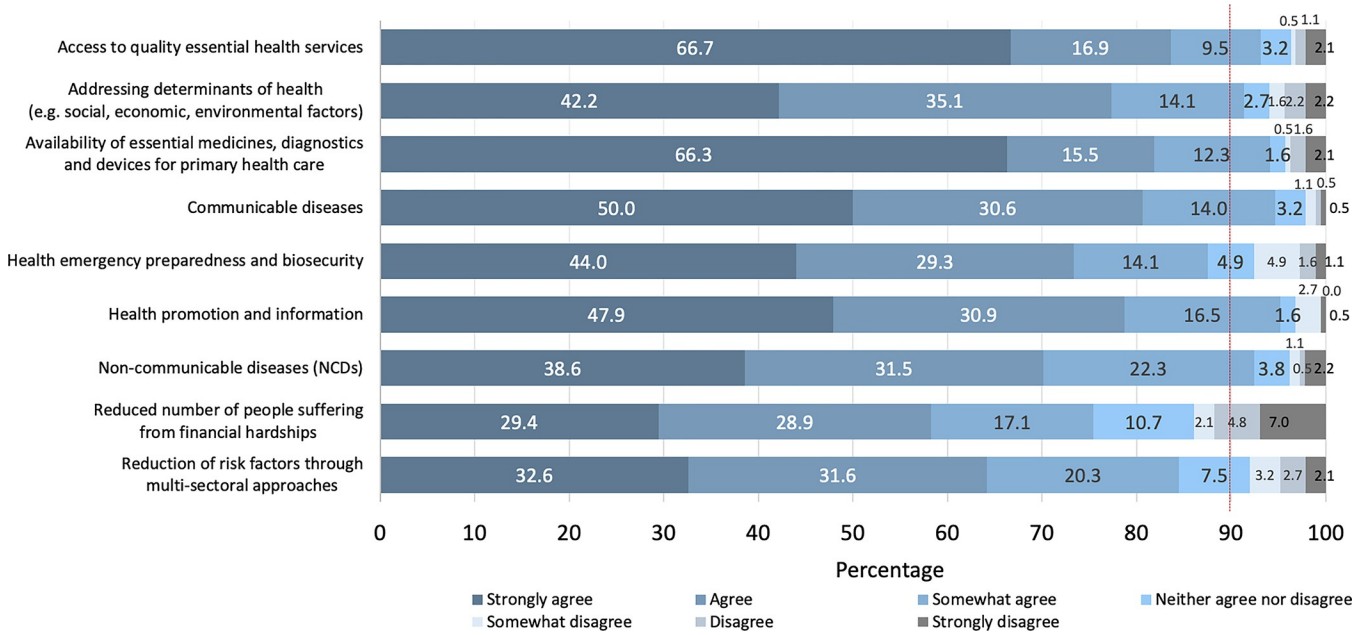

**Fig 3. Perception of the most essential health priority for Uganda by health care workers (*n* = 274).**

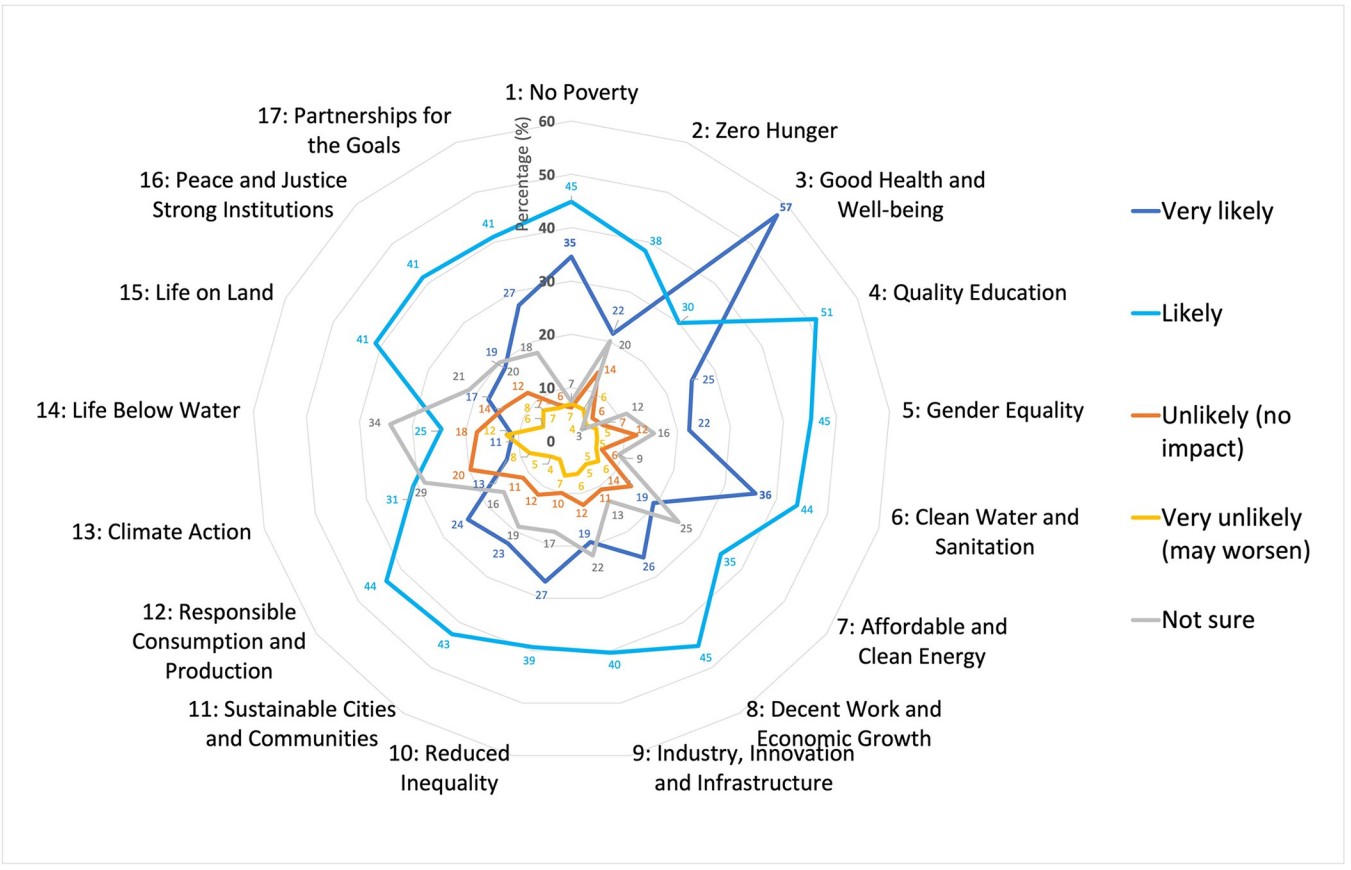

**Fig 4. Perceived impact of UHC on the 17 UN SDGs.** The radar chart displays the outcomes of the 5-point Likert scale, and the datapoints are shown as percentages, whereby the blue lines represent positive responses and the other colours (orange, yellow and grey) symbolise more negative or inconclusive responses. The angular axis (full circle) illustrates each of the SDGs and the radial axis (line of percentages) shows the frequency of responses as a percentage. The peaks show the most prominent choice or highest frequency of responses for each of the SDGs as a percentage.

Most respondents perceived a high level of urgency of UHC in Uganda, noting that it was the highest priority for health and should be implemented within one year.

**Impact of UHC on the 17 United Nations Sustainable Development Goals (UN SDGs).** More than half of all respondents reported that UHC would be 'very likely' to have an impact on Goal 3 of 'Good Health and Wellbeing' (57%) of the UN SDGs (Fig 4). UHC was further perceived to have a 'very likely' impact on two additional SDGs, namely Goal 6 on 'Clean Water and Sanitation' (36%) and Goal 1 on 'No Poverty' (35%). There was a more positive than negative sentiment on the perceived impact of UHC on the SDGs.

**Insights on the coronavirus disease 2019 (COVID-19) pandemic from open-ended responses.** Lessons learned from the COVID-19 pandemic included the need for emergency preparedness plans and response readiness appropriate to the country, strengthening health system infrastructure, medical supplies, equipment, and digitalisation, providing sufficient health sector funding for interventions, enhancing HCW remuneration, awareness creation, health promotion and education and the importance of infection prevention and control measures.

**Semi-structured in-depth interviews and focus group discussion (FGD).** Overall, 23 HCWs took part in the semi-structured interviews (Table 2) and eight HCWs participated in the focus group discussion (Table 3). Out of the total participants, 10 (32%) were female and

**Table 2. Selected socio-demographic data of health care workers comprising the semi-structured interview group (*n* = 23).**

| Count | Level | Institution | Type of organisation or facility | Sex |
|---|---|---|---|---|
| 1 | National | Government | Research Institute | Male |
| 2 | National | Government | National Referral Hospital | Male |
| 3 | National | Government | Health Centre III | Female |
| 4 | National | Government | Private Hospital | Male |
| 5 | National | Government | Laboratory | Male |
| 6 | National | Government | Health Centre (undisclosed number) | Female |
| 7 | National | Government | Laboratory | Male |
| 8 | National | Government | Laboratory | Male |
| 9 | National | Government | Research Institute | Male |
| 10 | National | Government | National Referral Hospital | Male |
| 11 | National | Government | General Hospital | Male |
| 12 | National | Government | Laboratory | Male |
| 13 | National | Government | Health Centre IV | Male |
| 14 | National | Government | National Referral Hospital | Male |
| 15 | National | Government | Public Health Institute | Male |
| 16 | National | Government | Research Institute | Male |
| 17 | National | Government | General Hospital | Male |
| 18 | National | Government | Research Institute | Male |
| 19 | National | Non-Governmental Organisation | Humanitarian Organisation | Male |
| 20 | National | Non-Governmental Organisation | Clinic | Male |
| 21 | National | Non-Governmental Organisation | International Organisation (Field Office) | Male |
| 22 | National | Non-Governmental Organisation | Network | Male |
| 23 | National | Private | Clinic | Male |

most (84%) worked in government institutions. Further demographic information beyond that listed in Tables 2 and 3 was not collected to safeguard the anonymity of respondents.

The thematic analysis gave rise to four overarching themes, namely in relation to: (1) organisation, (2) power, (3) communication, and (4) trust (Table 4). Some of the reported barriers to UHC are presented below, according to the six building blocks of health systems [4]. This WHO framework on financing, health workforce, information, leadership and governance, medical products, and service delivery encapsulates the basic functions for health system strengthening. Overall, there was general consensus on the lack of clarity of UHC and its limited public awareness. UHC was considered an important effort worth striving towards by participants, but challenges related to available strategies, implementation, accountability, and

**Table 3. Selected socio-demographic data of health care workers in the focus group discussion (*n* = 8).**

| Count | Level | Institution | Type of facility | Role | Profession | Sex |
|---|---|---|---|---|---|---|
| 1 | National | Government | Health Centre | Nursing Officer | Health care worker | Female |
| 2 | National | Government | Health Centre | Counsellor | Health care worker | Female |
| 3 | National | Government | Health Centre | Midwife | Health care worker | Female |
| 4 | National | Government | Health Centre | Midwife | Health care worker | Female |
| 5 | National | Government | Health Centre | Linkage Facilitator | Health care worker | Female |
| 6 | National | Government | Health Centre | Linkage Facilitator | Health care worker | Female |
| 7 | National | Government | Health Centre | Clinical Officer | Health care worker | Female |
| 8 | National | Government | Health Centre | Peer mother/ Counsellor | Health care worker | Female |

Table 4. Overview of key emerging themes from data on the perceptions and awareness of UHC among stakeholders.

| Main theme* | Six building blocks of health systems | Barriers (examples derived from data) | Key messages (and facilitators) |
|---|---|---|---|
| I–Organisation | Leadership and Governance | • Limited quality health infrastructure<br>• Lack of safe water access<br>• Power outages<br>• Poor physical accessibility of health facilities and referral system<br>• Lack of available services<br>• Procurement issues | 1. Health care workers valued through appropriate remuneration<br>2. Efficient cross-sectoral collaboration and cooperation<br>3. Harmonisation of objectives and efforts |
| | Health Workforce | • Overworked health professionals<br>• Understaffed health facilities<br>• Lack of professionalism impacting health services | |
| II–Power | Financing | • Budgeting issues<br>• Insufficient public financing for health<br>• Downplaying of activities to redivert funds<br>• Mishandling of funds and weak accountability measures | 1. Provision of financial resources and accountability of utilisation<br>2. Balance among stakeholders involved<br>3. Awareness of strategies and information power |
| III–Communication | Health Information | • Lack of community sensitisation and involvement in UHC strategies<br>• No apparent communication strategy<br>• Concept of UHC unclear<br>• Lack of grassroots level knowledge of UHC | 1. Sensitisation of concept and community inclusion<br>2. Empowerment of own entitlements<br>3. Accessible information on UHC for raising awareness |
| IV–Trust | Access to Medical Products | • Medicine stock outs<br>• Expired and sub-standard medications affecting care<br>• Lack of medical products and equipment | 1. Accountability and deliver on promises<br>2. Confidence in receiving intended services<br>3. Receive services needed of good quality |
| | Service Delivery | • Needed services not received by patient<br>• Lack of quality services and skilled personnel<br>• Access to services costly<br>• Incorrect perception of free health care services | |

*Note: The themes were underpinned by the six health system building blocks conceptualisation [4].

None of the FGD participants had heard of UHC prior to the discussion and participating in the survey, thus the guiding probes specific to UHC were re-formulated into questions related to health service access and provision, and financial implications in Uganda.

multisectoral collaboration were perceived as some of the roadblocks in this context. Beyond the organisational, political, and social contexts and processes related to UHC, the concept of trust emerged as the backbone of many issues discussed.

## I. Organisation

**i. Leadership and governance.** Most interview participants highlighted the significance of communication in multisectoral collaboration across organisations or in public-private partnerships (PPPs) and creating synergies to harmonise activities instead of duplicating efforts. Concerns were expressed in relation to limited quality health infrastructure and safe water access, medicine stock outs, power outages, and poor physical accessibility of health facilities. While some reported that the decentralised health system structure, from lower-level health facilities, i.e. health centre (HC) IIs, up to the General Hospital level, functioned well, others were more critical of the poor referral system, and limited hospitals with sufficient health personnel. Although supervision structures were in place, cases existed wherein presumed services were not availed in reality:

*"Where health centre III is. . . supposed to be having admission, but most of health centre IIIs in Uganda here, they don't have admission facilities. They have only services that can cater for*

*mothers. . .. which caters (for) maternity, yes, whereby a pregnant mother. . . go(es) for antenatal services for free of charge until delivery." (SSI06, female HCW)*

Several participants discussed procurement and funding issues related to supply chain management. Views of the National Medical Store (NMS), the sole supplier of essential medical supplies of all public health facilities in the country, were divided, with participants suggesting that supply did not meet the needs of a community. Participants noted that supplies would arrive too late or were unaffordable for patients, considering the high out-of-pocket (OOP) costs:

*". . .you see in health facility 'A', which I can't mention now, they have banned bunches of drugs, because they are expired. Now, the question is, does it mean the uptake for drugs is low? But the answer is that it is not that the uptake for drugs is low, it is not–the reason is that drugs have come late. . ." (SSI22, male HCW)*

Many (18 out of 23) interview participants described the value of UHC and political will needed for effective implementation of health interventions. It was noted that the implementation of the minimum comprehensive benefit package of health services, the Uganda National Minimum Health Care Package would need to be accompanied by adequate public funding of the health system. Several interview participants pointed out the lack of an overt effort by government to embrace UHC in health facilities, with some arguing that stating commitment at international level had become a *"political token"*. Interview participants were also critical of the tendency of overinvesting in curative services while underinvesting in preventive services.

**ii. Health workforce.** All participants echoed the omnipresent problems of overburdened and overworked health professionals, understaffed health facilities and poor remuneration. One participant described a scenario wherein services were decentralised to the sub-district level and new roles were given to an existing team of practitioners, who in turn were not able to commit to the new roles due to being overstretched by a surplus of tasks. Gaps in human resources compounded the issue, as it meant that fewer personnel conducted a greater number of tasks.

*"Because, in most cases, we are affected by seeing that the team is still very small and if it is small, we work tirelessly, and at the end of the day, they cannot make it perfect." (SSI08, male HCW)*

General frustrations of the health system discussed by participants included systemic issues that affected the work ethic of some professionals. One participant described lack of professionalism in health service availability:

*". . . someone wakes up and says, 'for me, I'll work up to 2pm' and then you find that–actually, most of them have already (left). By 2pm, they are telling you, 'Why didn't you come in the morning?'. . . you expect a health centre to work. . . day and overnight. If it has admissions, say if someone coming at that time is very sick and you're telling the patient, 'You're late'. Mm? Sickness has no time." (SSI12, male HCW)*

## II. Power

**i. Financing.** The importance of resource mobilisation and strengthening national domestic financing through improved public finance mechanisms was emphasised by most participants. One participant argued that as the country struggles to achieve middle-income status,

donors would begin to withdraw and thus equitable resource mobilisation and accountability, for instance through PPPs, would be critical. Public financing for UHC would need to be increased through raising or maintaining the tax base at sufficient levels.

> "*Because people are healthy, the tax base will improve and when the tax base improves, the GDP (Gross Domestic Product) of the country will improve and services and money will come back to the community.*" (SSI15, male HCW)

Corruption was considered another barrier to UHC, particularly the misappropriation of funds, unaccounted resources either through weak accountability or auditing mechanisms or gaps in planning, hampering the achievement of objectives. Reported examples of corrupt practices included bribery, diversion of project funds for personal reasons, and officials demanding additional reimbursement for attending an event despite receiving financial government support for transport. A few interview participants noted that embezzlement practices were difficult to disrupt, as they had taken root in the country with similar mentalities across generations. One interview participant stressed a related issue:

> "*Corruption. . . . When–me, I largely work in a sub-government–it's not a public service. But we normally work with public health system leaders. They normally want you to have a budget given to them for any activity you have. For them to manage, alright? So, if you do not have a budget, they downplay your activity. That's a profound challenge–*" (SSI03, male HCW)

### III. Communication

**i. Health information.**   Most participants expressed the importance of raising awareness among and involving communities in advocacy strategies for UHC and conveyed a critical gap in this area. Without the awareness and involvement of local people through community mobilisation, any efforts towards its implementation may be in vain. Examples of communication strategies included posters or using radio stations to disseminate information. Community sensitisation was one of the most important aspects discussed by most (27 out of 31) participants in supporting strategies for UHC. In bringing information related to UHC to local levels, participants emphasised the value of involving religious leaders who were able to influence attitudes of communities.

> "*But the leaders in all sectors, it may be cultural leaders, religious leaders also have a very great role in universal health coverage*" (SSI18, male HCW)

A negative sentiment towards the clarity of UHC arose among a few participants, with some describing UHC as being a *"vague"* concept that failed to reach the grassroots level of the local person. UHC was seen as a vehicle by donor organisations or NGOs to *"justify funding"*. An interview participant argued that the UHC concept did not impact the manner by which the government was doing business in terms of health care delivery, since enhancing access to health services at the last mile possible already existed within their health priorities prior to UHC.

> "*. . . the ideology of UHC–they need to make it a little bit pragmatic. It's so conceptual and so abstract, eh?*" (SSI12, male HCW)

## IV. Trust

**i. Access to medical products.** Repercussions of delays in medical supplies due to the pandemic were revealed in form of expired and sub-standard medications. In addition, attempting to combine delivery cycles in the supply of medicines posed further challenges in result of medicine stock outs juxtaposed with a lack of sufficient storage space. The need to improve upon poor ambulance and referral services and road infrastructure, as well as the importance of health system resilience and emergency preparedness re-emerged through the pandemic. One interview participant reported the challenge of prioritised COVID-19 medical supplies, such as vaccines, face masks, and hand sanitizers, within the NMS, at the expense of other essential medicines and products that were *"put on hold"* (SSI14, male HCW).

Another key challenge was the lack of provision of personal protective equipment (PPE) among HCWs, thus jeopardising their safety and exposing them to infection. Redundancies, task shifting and the increasing workload of HCWs due to the pandemic also carried mental health consequences, such as burn-out, compounding the problems that HCWs had to manage. It was also reported that a number of health professionals died from COVID-19 and other comorbidities throughout this period (SSI23, male HCW). One participant who fell sick with COVID-19 expressed he was prescribed forced bed rest, urged by another health professional, as the participant *"was going to die from the community"* due to exhaustion through overworking (SSI11, male HCW).

**ii. Service delivery.** Most (29 out of 31) participants discussed the persisting challenge of health service accessibility and availability barriers to utilisation of vulnerable people in rural communities and remote areas. It was emphasised that medical supplies at health facilities could not meet the demands of the communities (SSI15, male HCW). The trust of an individual in the health system was reported to be greatly affected by those who faced negative experiences, such as being turned down at health facilities and returning home without having received expected health services (SSI10, male HCW). Poor logistics, such as early closing times of village health centres or a lack of quality services and skilled personnel also impacted individual health seeking behaviours. One participant described that the trust individuals placed on them as being a motivational factor in their work (SSI23, male HCW).

Another participant described that access to health care was contingent on the residence and financial status of an individual:

*"Access to health care majorly in my country, it depends on the region where you're coming from and your financial status. Though government has its facilities and they are not so well stocked from my side. . . . health care providers, their number is not sufficiently handling the relation sometimes. And also the facilities are always constrained in terms of structure, in terms of logistics."* (SSI17, male HCW)

The focus group discussed the issue of stock outs of essential medicines and supplies, detailing that often they had to wait for two months for refills in their facility. One discussant criticised that despite a rapidly increasing population in the country, the health budget remained firmly constant. There was consensus by the group on the incorrect perception of free health services in the country:

*"In our setting (health centre 4), it's a government facility, where clients are not supposed to pay. But I would say at the end of the day, I would say they end up [more] spending (more money than anticipated)."* (FGD1, female HCW)

## Discussion

Overall, the stakeholder analysis provided further knowledge on the perceptions and importance of achieving UHC in Uganda by HCWs. Results suggested a perceived high level of importance attributed to UHC. Increased financing and strong political will were considered the most important factors in achieving UHC. Concurrently, limited financing and the political economy were also reported as the main barriers to UHC by the online survey respondents. Key messages of the interviews and FGD covered four themes, namely: *(1) trust*, *(2) communication*, *(3) power*, *and (4) organisation*. For *trust*, participants highlighted the importance of accountability and receiving needed services of good quality. *Communication* involved community sensitisation of UHC and accessible information for raising awareness. *Power* included the provision of financial resources and accountability of utilisation, and central to *organisation* was valuing HCWs through appropriate remuneration, efficient multisectoral collaboration, and a harmonisation of efforts.

Participants highlighted that UHC strategies ought to include accountability mechanisms for effective implementation leading to impact, such as through enhancing monitoring and evaluation (M&E) and supervision practices or establishing distinct focal points for UHC with a financial contingency. Most participants were not aware of the national UHC Roadmap, which could have impacted some of their views. This lack of awareness could have been partially due to the publication timing, as it coincided with the emergence of the global COVID-19 pandemic. Concerning inadequate funding within the health sector, participants implied skewed government priorities towards the security sector, with health-related issues not being a priority. Evidence from other countries in the region, such as Botswana, Eswatini, the Seychelles, or South Africa, within the literature have highlighted the benefits of high political commitment and accountability, increasing domestic financing for UHC, investing in PHC and incremental changes for the advancement of population health [33–38]. Finally, participants noted that UHC strategies should help tackle the root causes of poverty and incorporate local community needs for health systems strengthening and sustainability.

This analysis also sheds light on the influences of power dynamics among different stakeholders in terms of the level of knowledge of UHC. Topp and colleagues argued that more research is needed on power in the field of health policy and systems and this empirical research aimed to provide further insight into this knowledge gap [39]. Ssennyonjo [40] highlighted the concept of *ideational power*, its influence in form of groupthink in political environments and the bidirectional influence of structure and agency in health financing reforms in Uganda. In particular, the influence of politics and power with regard to health financing was highlighted among participants who discussed opposing forces, such as wealthier individuals who provide a large part of the tax base in public financing. Concepts of power also emerged in the online survey, where respondents described power plays in political environments to the detriment of programmes and a disconnect between top-level district officials and lower-level staff exacerbating policy-to-implementation gaps, echoed by the literature [14, 41]. This disconnect may propagate into a further imbalance of knowledge with key stakeholders not involved in the main discourse, a notion that was described in Foucault's influential work on the intricate interplay of knowledge and power [42].

In view of the projected global shortage of 18 million HCWs by 2030 [43], findings from this analysis touched on neglected frontline health workers confronted with overburdened and understaffed health facilities, referral system inefficiencies and a lack of proper remuneration. Participants agreed that confidence in receiving intended services of quality was central to the development of trust. This finding was corroborated in the literature, whereby a systematic review of stockouts, a complete absence or unavailability of medical products, including

medicines and vaccines, among community health workers in low- and middle-income countries (LMICs) suggested an increase in stockout rates from 2016 to 2021, noting that procurement and distribution issues, such as governance and coordination problems, were linked with trust issues among the population [44, 45]. Although subtleties related to the level of trust in health settings is difficult to measure, earlier research on the conceptualisation of trust has suggested a relationship between trust and knowledge, in the sense that trust may promote timely information sharing [46, 47].

Finally, participants highlighted the importance of disseminating clear information on UHC in order to raise awareness. Particularly, the perception that UHC was a vague concept, which was not easily understood, and failed to reach the grassroots level was a concern shared by many participants. In communicating information, all levels need to be represented for a complete transfer of knowledge, to avoid information loss and the clear, correct concept of UHC being restricted to a small group of experts [48, 49]. This is reinforced within the literature, whereby a cross-sectional online survey of 326 nurses in Hong Kong conducted by Tung et al. 2016 revealed that nurses being relatively indifferent to health policies created knowledge gaps that affected the success of UHC initiatives [15]. In addition, participants of this study emphasised that a *bottom-up approach* is critical, thus involving expertise of different localities is needed to reach the end goal.

## Limitations

A key limitation involved reaching relevant stakeholders. The low number of online survey responses (2.3% response rate), possibly due to disinterest in participation, lack of time, or issues in obtaining correct contact details, were addressed through further snowballing methods. Employing purposive or snowball sampling meant that the final sample was not representative of the target population of stakeholders in UHC in Uganda. Additionally, issues of representativeness and generalisability to other low-income countries need to be considered. The low response rate of the survey limits the generalisability. Further exploration of the predictors or factors influencing (i) knowledge of UHC, and (ii) awareness towards UHC through regression analyses would have been interesting, but the data collected was not detailed enough for this to be undertaken. Nevertheless, the benefits of this approach during the pandemic included facilitating the reach of individuals in a cost-effective and time-saving manner. Efforts were made to increase participation and response rates based on recommendations from a previous study of an e-mail survey of HCWs in Uganda on the importance of point-of-care diagnostics to tackle antimicrobial resistance (AMR) [50].

Furthermore, contrary to the procedure for the online survey, due to time constraints, the qualitative interviews and FGD topic guides were reviewed by Ugandan based team members, not formally piloted. Biases, mistranslations, and a lack of contextual adaptation might have ensued during the ad hoc translations for the one non-English speaking participant. A pilot test of the guide could have enabled a more contextually and culturally adapted tool. However, piloting was not possible due to the constraints of the timeframe that the pandemic created. English is the national language of Uganda, and all health care worker documentation and trainings are conducted in English. Most qualitative interviews in Kampala with health care workers will be in English. However, sometimes health care workers are more comfortable in their local language and inclusion of a non-English FGD may have been useful.

Response bias was accounted for through the development of an anonymous online survey and by acknowledging potentially skewed responses throughout the interpretation stages, as individuals in favour of UHC would have responded more positively. Caution was exerted with interpretations of transcripts, safeguarding the context, tone, and non-verbal cues, where

feasible, while determining their meaning. However, data analysis was solely conducted by one researcher, which could lead to researcher and cognitive biases. Possible subjective bias was addressed through self-critical, reflexive, and an open approach for discussion by the research team at multiple stages of the research.

Despite these limitations, this analysis attempted to unravel the complexity of UHC and stakeholder perceptions of this in Uganda. Harmonisation of interests and efforts in multisectoral collaboration were among the main reported barriers to UHC. Some of these challenges resonated with evidence from the literature, where working in silos resulted in duplicated efforts and limited policy implementation [12, 51]. Further challenges related to collaboration, accountability and management were disclosed, such as the duplication of efforts, conflicting policies, and a need for alignment of interests. Addressing such challenges while preserving project sustainability was perceived as crucial if UHC were to be realised in the country.

## Public health policy implications and recommendations

Most of the key messages delineated from the findings have implications for global health policy and in LMICs. In terms of multisectoral collaboration practices and a systems thinking approach required for UHC, power and trust were considered key constituents. To safeguard trust, robust accountability mechanisms, a guarantee of fulfilment of promises or objectives and, most importantly, transparency and honesty are important considerations for the delivery of any service or strategy. In the same regard, accessible and continuous communication at various levels, considering power dynamics between local communities and national entities, remains pivotal to achieve transparency, sensitisation, and a greater awareness of UHC principles.

Participants in this study identified the absence of a comprehensive and well-defined communication strategy for UHC to enable more widespread awareness. Such a communication strategy could benefit from the support of an established technical working group in implementing the already defined UHC Roadmap. Equally, one or more focal points for UHC to help guide developments, provide a main access point for the different levels of stakeholders, and actively promote the dialogue among these key stakeholders could help move the strategies forward. The focal point would also be instrumental in implementing the communication strategy, in quality control and disseminating related public health messaging for UHC.

Finally, despite the multitude of different stakeholders involved in UHC and the enormity of this collaboration, it remains imperative to consider and value those individuals providing the muscle of health service delivery, the HCWs. It is important to be mindful of personal motivational factors of individuals and incorporating these into strategies for improved ownership, empowerment, and implementation. Protecting and nurturing the 'heart' or reason behind the work HCWs perform could help improve relationships and satisfaction in the working environment. To develop adequate, competent, and motivated personnel who are equitably distributed across health facilities in the country, appropriate remuneration and sufficient financing need to be accounted for.

## Conclusion

There was a nuanced understanding of UHC as a concept even among the key informants, with a general knowledge of its principles despite a lack of awareness of central strategic documents, such as the more recent ministerial UHC Roadmap. This stakeholder analysis provided further insights into the perceptions of HCWs on UHC. A common issue identified by participants related to communication, particularly with access to information remaining a problem. As much as a conducive political environment, strong political will and incorporating the multisectoral nature of UHC are important, communication inclusive of all levels, including the

grassroots and community level, is imperative in sensitisation of UHC and if health outcomes are to be improved for the wider society.

Despite the multitude of multisectoral stakeholders of UHC, it remains imperative to value those individuals providing the muscle of health service delivery, the HCWs. To develop adequate, competent, and motivated personnel, equitably distributed across health facilities in the country, appropriate remuneration is pivotal. Considering that "sickness has no time" and thus can occur at any moment, strong health care systems can make the difference between life and death.

## Supporting information

**S1 Appendix. Online UHC survey–background file for Qualtrics<sup>XM</sup>.**
(PDF)

**S1 Table. UHC-related responses on knowledge and awareness by health care workers from the online survey ($n$ = 274).**
(PDF)

**S2 Table. Selected characteristics and UHC-related responses of health care workers by knowledge of UHC ($n$ = 274).**
(PDF)

**S3 Table. Selected characteristics and UHC-related responses of health care workers by rural and urban locations of the health facility ($n$ = 274).**
(PDF)

## Acknowledgments

We are grateful for the participation of all study participants and the colleagues in The Academy for Health Innovation at the Infectious Diseases Institute, Makerere University, in Kampala, Uganda. This study is part of a PhD thesis supported by the University of Cambridge.

## Author Contributions

**Conceptualization:** Susan C. Ifeagwu, Carol Brayne, Rosalind Parkes-Ratanshi.

**Data curation:** Susan C. Ifeagwu.

**Formal analysis:** Susan C. Ifeagwu, Stephen Ojiambo Wandera, Suzanne N. Kiwanuka, Carol Brayne, Rosalind Parkes-Ratanshi.

**Investigation:** Susan C. Ifeagwu.

**Methodology:** Susan C. Ifeagwu, Suzan Nakkazi, Joshua Beinomugisha, Stephen Ojiambo Wandera, Carol Brayne, Rosalind Parkes-Ratanshi.

**Project administration:** Susan C. Ifeagwu, Ruth Nakaboga Kikonyogo.

**Resources:** Susan C. Ifeagwu.

**Software:** Susan C. Ifeagwu.

**Supervision:** Carol Brayne, Rosalind Parkes-Ratanshi.

**Validation:** Susan C. Ifeagwu.

**Visualization:** Susan C. Ifeagwu.

**Writing – original draft:** Susan C. Ifeagwu.

**Writing – review & editing:** Susan C. Ifeagwu, Stephen Ojiambo Wandera, Suzanne N. Kiwanuka, Rachel King, Tine Van Bortel, Carol Brayne, Rosalind Parkes-Ratanshi.

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
