## [Decision Letter · Decision Letter 0]

10 Mar 2024

PONE-D-23-38248"Sickness has no time": Awareness and perceptions of health care workers on universal health coverage in UgandaPLOS ONE

Dear Dr. Ifeagwu,

Thank you for submitting your manuscript to PLOS ONE. After careful consideration, we feel that it has merit but does not fully meet PLOS ONE’s publication criteria as it currently stands. Therefore, we invite you to submit a revised version of the manuscript that addresses the points raised during the review process.

**ACADEMIC EDITOR: Please insert comments here and delete this placeholder text when finished.** Be sure to:Editor commentsPlease address the reviewers’ comments, including adding further consideration on methods and discussion. You will see that one of the reviewers has suggested improvements on methods and the other one ways of discussing the results. To ensure the Editor and Reviewers will be able to recommend that your revised manuscript is accepted, please pay careful attention to each of the comments that have been shared with you. Please also consider improving the concluding remarks and policy implication of your work. This way we can avoid future rounds of clarifications and revisions, moving swiftly to a decision.Please ensure that your decision is justified on PLOS ONE’s publication criteria and not, for example, on novelty or perceived impact.

We look forward to receiving your revised manuscript.

Kind regards,

Dr Patrick Christian Ilboudo

Academic Editor

PLOS ONE

Journal Requirements:

2. Please remove your figures from within your manuscript file, leaving only the individual TIFF/EPS image files, uploaded separately. These will be automatically included in the reviewers’ PDF.

Additional Editor Comments:

As per one of the reviewers comments, please ensure that the manuscript has been proofread.

Reviewers' comments:

Reviewer's Responses to Questions

**Comments to the Author**

1. Is the manuscript technically sound, and do the data support the conclusions?

Reviewer #1: Yes

Reviewer #2: No

2. Has the statistical analysis been performed appropriately and rigorously? 

Reviewer #1: No

Reviewer #2: Yes

3. Have the authors made all data underlying the findings in their manuscript fully available?

Reviewer #1: Yes

Reviewer #2: Yes

4. Is the manuscript presented in an intelligible fashion and written in standard English?

Reviewer #1: No

Reviewer #2: Yes

5. Review Comments to the Author

Reviewer #1: Overall Comment

This is an interesting article examining the perceptions towards the priorities and importance of UHC among healthcare workers in Uganda. I think this is an important topic for policymakers, researchers and practitioners working on health systems strengthening. However, there are some major shortcomings in several sections of the paper, in particular, how the information is presented as well as discussed. Below are the detailed comments where I point specifically to the limitations, as well as suggestions to improve them.

Specific Comments

1) In the second paragraph of ‘study design’ sub-section, there is description about the online survey but one key information is lacking: how was the survey designed? Was it based on a prior instrument that has been deployed somewhere else or any published study? Or was it designed from scratch by the researchers? Were the questions validated and adapted to the Ugandan cultural context? I think it would be important to incorporate these information in the study design sub-section as well.

2) Table 1 currently appears to be rather messy and overloaded with information. I think the figures can be presented more succinctly and clearly. I think there is no need to spell out categories for ‘other’ and details for ‘other’ in the table. Rather, these information can be placed as notes under the table.

3) The authors should explain how the radar chart (Figure 3) is interpreted better. I am somewhat confused by the information presented here. How is goal 3 (health and well-being) perceived as very likely to be impacted by UHC by more than half of the respondents? Where does the 57% come from? It is not very intuitive if I look at the radar chart. How about the percentages for Goal 6 and Goal 1? Again, I cannot see these information from the radar chart. Perhaps it is better to present the information in a table format with clear figures instead of a vague radar chart, unless the authors describe clearly how should this radar chart be interpreted by the readers.

4) Table 2 seems to appear from nowhere. I am not sure if there is any value presenting the findings in this manner. Perhaps there is no need to present these information. Even if the authors would like to, I would suggest making them as narratives/texts.

5) I think Table 3 can be presented in a different manner. First, I do not think there the description column is redundant as each of the six building blocks is already quite clear and intuitive to the readers. What can go to the table, instead, would be a summary of the facilitators for and barriers to each of the six building blocks. What the authors can do here is to summarize the narratives in the main text as bullet points here.

6) In addition to all the findings presented in the article, I find that there might be another interesting analysis which the authors could explore. What are the predictors or factors influencing (i) knowledge of UHC, and (ii) awareness towards UHC? Can regression analyses be performed to tease out what might be some of the demographic, social and institutional factors influencing the above two dependent variables (knowledge and awareness towards UHC)?

7) Discussion section (P23 L470-472), the authors wrote “Financing and political will were considered the most important factors in achieving UHC, and conversely, financing and the political economy were reported as the main barriers.” This sentence is unclear and vague, I do not quite understand what the authors mean here. Please revise.

8) The discussion section is too thin. I think there needs to be deeper reflection on the following points:

a. implications of findings to the UHC agenda and UHC implementation in Uganda.

b. What do the findings from this study mean for other low- and middle-income countries (LMIC)? In other words, what can other LMICs learn from your findings?

c. Generalisability (external validity) of your findings needs to be discussed as well.

d. What would be some of the policy recommendations in lieu of the findings.

9) This article needs to be edited more carefully. I spotted some grammatical mistakes throughout the paper, just to highlight a few more obvious one:-

a. P.4 L97 ‘focussed’ should be ‘focused’.

b. P.5L140 ‘The final online survey was comprised of an introductory section…’ should be ‘The final online survey comprised an introductory section…”

c. P.9 L239-240, the authors wrote ‘Most respondents were based in an urban setting, as opposed to a rural one, which was derived from the health facility location’. This sentence is very convoluted, can be simplified as ‘most respondents were based in an urban setting’. What do you mean by ‘derived from the health facility location’? Please clarify and revise.

d. P.24 L499: Shouldn’t ‘considering these limitations’ be ‘despite these limitations’?

Reviewer #2: The topic is interesting because UHC is a concept that need to be understood by the implementers. The way the implementers will understand UHC will affect its implementation.

However there are several clarifications that authors need to provide mainly in the method and discussion sections.

Title

the title needs to be reformulate to reflect the content. I suggest a title : Awareness and perceptions of health care workers on universal health coverage in Uganda

Introduction

From Line 102 to 106. The authors presented the results of a study in Asia. I suggest to present results from study in Africa related to UHC perceptions such as Koon AD, et al 2017.

Method

Study design

The use of mix method is not clear. What is the sequence of utlization and how each of these studies are linked ?

The online survey was sent to the entire sub-Saharan African region (line 151) but the scope of the study was uganda, why did you select these sample from sub saharan african region. How did the author estimate the sample size for the online survey ?

They sent too many email to many individual, how did they insure that there was not double entry.

About the sampling of the in-depth semi-structured key informant interviews, from line 164 to 165 authors stated that « the in-depth semi-structured key informant interviews were selected through purposive ». That is in contracdiction with what they stated on line 173 « Further interviewees were identified through the snowballing technique to attempt to expand the breadth of informants included. » Please explain how the interviewees were selected.

How did the authors check the saturation ? Please explain.

Why did you use semi structured key informant interviews and focus group discussion. What was the add value of each technique ? Especially because the questions for the FGD were drew from the online survey like the individual semi structured interviews. In adition there was only 1 FGD. Please explain why did you conduct only one FGD ?

The population study was not clearly defined for each survey . Also the selection was not clear

Results

-How many individual interviews were conducted ? how many parcipants did you have in FGD ? What were their background ? Even if youd id not collect their name , information related to their professionnal category, organization, level of facility could be collected.

-On line 323, the authors stated that « None of the FGD participants had heard of UHC prior to the discussion, ». That is surprising because the praticipants for the FGD were drawn from the online survey. How could you explain this situation ?

The categorization of the six building blocks into the main themes is not clear because some themes such as trust or communication are backbone of challenges in the health system. Why did you choose to categorise like this ? I suggest to revise the presentation by looking at the main theme for each block.

Discussion

The discussion is confusing and not reflect the results. I suggest a deep revision of this section taking into account the results and interpretations. To reinforce the discussion, a literature review will be helpful on the different themes and results.

6. PLOS authors have the option to publish the peer review history of their article (what does this mean?). If published, this will include your full peer review and any attached files.

Reviewer #1: No

Reviewer #2: No

---

## [Author Response · Author response to Decision Letter 0]

4 May 2024

Point-by-Point Response to Reviewers

Editor comments:

Please address the reviewers’ comments, including adding further consideration on methods and discussion. You will see that one of the reviewers has suggested improvements on methods and the other one ways of discussing the results. 

To ensure the Editor and Reviewers will be able to recommend that your revised manuscript is accepted, please pay careful attention to each of the comments that have been shared with you. Please also consider improving the concluding remarks and policy implication of your work. This way we can avoid future rounds of clarifications and revisions, moving swiftly to a decision.

2. Please remove your figures from within your manuscript file, leaving only the individual TIFF/EPS image files, uploaded separately. These will be automatically included in the reviewers’ PDF.

As per one of the reviewers’ comments, please ensure that the manuscript has been proofread.

Response and Corrections:

Thank you for this opportunity to revise the manuscript and address the reviewers’ comments. As advised, each of the comments have been given due attention and addressed within the revised manuscript. All figures have also been removed, the manuscript has been proofread, and corrections have been made. 

Finally, all line and page numbers correspond to those in the ‘Revised Manuscript with Track Changes’ version.

Please find all revisions and accompanying explanations provided below.

Reviewer 1:

Overall Comment: This is an interesting article examining the perceptions towards the priorities and importance of UHC among healthcare workers in Uganda. I think this is an important topic for policymakers, researchers and practitioners working on health systems strengthening. However, there are some major shortcomings in several sections of the paper, in particular, how the information is presented as well as discussed. Below are the detailed comments where I point specifically to the limitations, as well as suggestions to improve them.

Response and Corrections:

Thank you very much for your insightful and constructive review, and for taking the time to critically examine our manuscript. We very much appreciate your suggestions and have incorporated the specific comments as described in the following sections. Please find below all corrections made.

Specific Comment 1:

1) In the second paragraph of ‘study design’ sub-section, there is description about the online survey but one key information is lacking: how was the survey designed? Was it based on a prior instrument that has been deployed somewhere else or any published study? Or was it designed from scratch by the researchers? Were the questions validated and adapted to the Ugandan cultural context? I think it would be important to incorporate these information in the study design sub-section as well.

Response and Correction 1:

The survey was designed from scratch and questions were based around the implementation strategy from the WHO Consultative Group on Equity and UHC. This has been clarified in the ‘study design’ sub-section through the following additions:

• Lines 159-161, page 5: “An original user-friendly, online email survey was developed by the researchers, to examine stakeholders’ perceptions of UHC and enhance survey accessibility [15, 17-19].”

• Lines 162-164, page 5: “No relevant prior instrument was found; thus, the online survey was developed by the researchers specifically to answer the specific research questions.”

The aim of the survey was to be general and user friendly enough for a wider group of respondents, as the survey was disseminated globally to international stakeholders. However, the interviews and focus group discussion were more culturally specific, and whilst not formally piloted, the FGD was shared with and supported by Ugandan based researchers. This issue was further reflected in the limitations section, as shown below:

• Lines 834-843, page 28: “Furthermore, contrary to the procedure for the online survey, due to time constraints, the qualitative interviews and FGD topic guides were reviewed by Ugandan based team members, not formally piloted. Biases, mistranslations, and a lack of contextual adaptation might have ensued during the ad hoc translations for the one non-English speaking participant. A pilot test of the guide could have enabled a more contextually and culturally adapted tool. However, piloting was not possible due to the constraints of the timeframe that the pandemic created. English is the national language of Uganda, and all health care worker documentation and trainings are conducted in English. Most qualitative interviews in Kampala with health care workers will be in English. However, sometimes health care workers are more comfortable in their local language and inclusion of a non-English FGD may have been useful.”

Specific Comment 2:

2) Table 1 currently appears to be rather messy and overloaded with information. I think the figures can be presented more succinctly and clearly. I think there is no need to spell out categories for ‘other’ and details for ‘other’ in the table. Rather, these information can be placed as notes under the table.

Response and Correction 2:

Thank you very much for the helpful idea. This suggestion has been taken on board, categories for ‘other’ have been moved to the notes section under the table, and Table 1 (lines 340-349, page 12) has now been presented in a more succinct way.

Specific Comment 3:

3) The authors should explain how the radar chart (Figure 3) is interpreted better. I am somewhat confused by the information presented here. How is goal 3 (health and well-being) perceived as very likely to be impacted by UHC by more than half of the respondents? Where does the 57% come from? It is not very intuitive if I look at the radar chart. How about the percentages for Goal 6 and Goal 1? Again, I cannot see these information from the radar chart. Perhaps it is better to present the information in a table format with clear figures instead of a vague radar chart, unless the authors describe clearly how should this radar chart be interpreted by the readers.

Response and Correction 3:

The radar chart has been amended to reflect all percentages and provide the data labels. Please find below the revised Figure 3.

Fig 3. Perceived impact of UHC on the 17 UN SDGs.

An additional explanation has been included within the figure caption to help with the interpretation of the radar chart:

• Lines 398-403, page 14: “Fig 3. Perceived impact of UHC on the 17 UN SDGs. The radar chart displays the outcomes of the 5-point Likert scale, and the datapoints are shown as percentages, whereby the blue lines represent positive responses and the other colours (orange, yellow and grey) symbolise more negative or inconclusive responses. The angular axis (full circle) illustrates each of the SDGs and the radial axis (line of percentages) shows the frequency of responses as a percentage. The peaks show the most prominent choice or highest frequency of responses for each of the SDGs as a percentage.”

This figure was included to provide a graphical presentation of the data and to avoid the use of too many tables.

Specific Comment 4:

4) Table 2 seems to appear from nowhere. I am not sure if there is any value presenting the findings in this manner. Perhaps there is no need to present these information. Even if the authors would like to, I would suggest making them as narratives/texts.

Response and Correction 4:

Thank you for this suggestion. Table 2 has now been deleted, as recommended. Instead a few of these findings were integrated into the results section, as shown below:

• Lines 407-413, page 15: 

“Insights on the coronavirus disease 2019 (COVID-19) pandemic from open-ended responses:

Lessons learned from the COVID-19 pandemic included the need for emergency preparedness plans and response readiness appropriate to the country, strengthening health system infrastructure, medical supplies, equipment, and digitalisation, providing sufficient health sector funding for interventions, enhancing HCW remuneration, awareness creation, health promotion and education and the importance of infection prevention and control measures.”

Other key findings from the former table were considered redundant and therefore deleted entirely.

Specific Comment 5:

5) I think Table 3 can be presented in a different manner. First, I do not think there the description column is redundant as each of the six building blocks is already quite clear and intuitive to the readers. What can go to the table, instead, would be a summary of the facilitators for and barriers to each of the six building blocks. What the authors can do here is to summarize the narratives in the main text as bullet points here.

Response and Correction 5:

Table 3 (now Table 4) has been revised according to the suggestion. The description column was deleted and replaced by two new columns: a summary of the barriers and key messages (facilitators), summarised by the narratives presented in the main text.

Revised Table 4 (formerly Table 3) on lines 467-478, pages 17-18, is shown below:

Table 4. Overview of key emerging themes from data on the perceptions and awareness of UHC among stakeholders

Main theme* Six building blocks of health systems Barriers (examples derived from data) Key messages (and facilitators)

I – Organisation 

 Leadership and Governance • Limited quality health infrastructure

• Lack of safe water access

• Power outages

• Poor physical accessibility of health facilities and referral system

• Lack of available services

• Procurement issues 1. Health care workers valued through appropriate remuneration

2. Efficient cross-sectoral collaboration and cooperation

3. Harmonisation of objectives and efforts

 Health Workforce • Overworked health professionals

• Understaffed health facilities

• Lack of professionalism impacting health services

II – Power Financing • Budgeting issues

• Insufficient public financing for health

• Downplaying of activities to redivert funds

• Mishandling of funds and weak accountability measures

 1. Provision of financial resources and accountability of utilisation

2. Balance among stakeholders involved

3. Awareness of strategies and information power

III – Communication Health Information • Lack of community sensitisation and involvement in UHC strategies

• No apparent communication strategy

• Concept of UHC unclear

• Lack of grassroots level knowledge of UHC

 1. Sensitisation of concept and community inclusion

2. Empowerment of own entitlements

3. Accessible information on UHC for raising awareness

IV – Trust Access to Medical Products

 • Medicine stock outs

• Expired and sub-standard medications affecting care

• Lack of medical products and equipment

 1. Accountability and deliver on promises

2. Confidence in receiving intended services

3. Receive services needed of good quality

 Service Delivery

 • Needed services not received by patient

• Lack of quality services and skilled personnel

• Access to services costly

• Incorrect perception of free health care services

*Note: The themes were underpinned by the six health system building blocks conceptualisation [4].

Specific Comment 6:

6) In addition to all the findings presented in the article, I find that there might be another interesting analysis which the authors could explore. What are the predictors or factors influencing (i) knowledge of UHC, and (ii) awareness towards UHC? Can regression analyses be performed to tease out what might be some of the demographic, social and institutional factors influencing the above two dependent variables (knowledge and awareness towards UHC)?

Response and Correction 6:

Thank you for this interesting idea for further analyses. While this would be a fascinating angle to explore, this idea is beyond the scope of the research objective and would have required a different approach, including exploring the literature on predictive analysis and related factors, to ensure validity of findings. In addition, the dataset was considered not extensive enough to conduct a robust regression analysis as suggested. We have added this to the limitations (lines 825-828), in the following way:

• Lines 825-828, page 28: “The low response rate of the survey limits the generalisability. Further exploration of the predictors or factors influencing (i) knowledge of UHC, and (ii) awareness towards UHC through regression analyses would have been interesting, but the data collected was not detailed enough for this to be undertaken.”

Specific Comment 7:

7) Discussion section (P23 L470-472), the authors wrote “Financing and political will were considered the most important factors in achieving UHC, and conversely, financing and the political economy were reported as the main barriers.” This sentence is unclear and vague, I do not quite understand what the authors mean here. Please revise.

Response and Correction 7:

The sentence has been revised to provide more clarity. The following changes have been made:

• Lines 680-682, page 25: “Increased financing and strong political will were considered the most important factors in achieving UHC. Concurrently, limited financing and the political economy were also reported as the main barriers to UHC by the online survey respondents.”

Specific Comment 8:

8) The discussion section is too thin. I think there needs to be deeper reflection on the following points:

a. implications of findings to the UHC agenda and UHC implementation in Uganda.

b. What do the findings from this study mean for other low- and middle-income countries (LMIC)? In other words, what can other LMICs learn from your findings?

c. Generalisability (external validity) of your findings needs to be discussed as well.

d. What would be some of the policy recommendations in lieu of the findings.

Response and Correction 8:

Thank you very much for this helpful comment. The discussion section has been revised and includes more information on the points raised within the comment. Moreover, the limitations section has been separated out and included under a new sub-heading called ‘Limitations’ (lines 818-862, pages 28-29).

A new section on the public health policy implications and recommendations has been included at the end of the discussion to address comments ‘a’, ‘b’, and ‘d’. More specifically, the following has been incorporated:

• Lines 864-893, pages 29-31: 

“Public health policy implications and recommendations:

Most of the key messages delineated from the findings have implications for global health policy and in LMICs. In terms of multisectoral collaboration practices and a systems thinking approach required for UHC, power and trust were considered key constituents. To safeguard trust, robust accountability mechanisms, a guarantee of fulfilment of promises or objectives and, most importantly, transparency and honesty are important considerations for the delivery of any service or strategy. In the same regard, accessible and continuous communication at various levels, considering power dynamics between local communities and national entities, remains pivotal to achieve transparency, sensitisation, and a greater awareness of UHC principles.

Participants in this study identified the absence of a comprehensive and well-defined communication strategy for UHC to enable more widespread awareness. Such a communication strategy could benefit from the support of an established technical working group in implementing the already defined UHC Roadmap. Equally, one or more focal points for UHC in order to help guide developments, provide a main access point for the different levels of stakeholders, and actively promote the

---

## [Decision Letter · Decision Letter 1]

26 Jun 2024

"Sickness has no time": Awareness and perceptions of health care workers on universal health coverage in Uganda

PONE-D-23-38248R1

Dear Dr. Ifeagwu,

We’re pleased to inform you that your manuscript has been judged scientifically suitable for publication and will be formally accepted for publication once it meets all outstanding technical requirements.

Kind regards,

Patrick Christian Ilboudo

Academic Editor

PLOS ONE

Additional Editor Comments (optional):

The authors have adequately addressed all the comments raised in a previous round of review by reviewer, therefore qualifying the paper for publication.

Reviewers' comments:

Reviewer's Responses to Questions

**Comments to the Author**

1. If the authors have adequately addressed your comments raised in a previous round of review and you feel that this manuscript is now acceptable for publication, you may indicate that here to bypass the “Comments to the Author” section, enter your conflict of interest statement in the “Confidential to Editor” section, and submit your "Accept" recommendation.

Reviewer #2: All comments have been addressed

2. Is the manuscript technically sound, and do the data support the conclusions?

Reviewer #2: Yes

3. Has the statistical analysis been performed appropriately and rigorously? 

Reviewer #2: N/A

4. Have the authors made all data underlying the findings in their manuscript fully available?

Reviewer #2: Yes

5. Is the manuscript presented in an intelligible fashion and written in standard English?

Reviewer #2: Yes

6. Review Comments to the Author

Reviewer #2: We are satisfied with the responses and revised version. We recommend that the manuscript will be accepted

7. PLOS authors have the option to publish the peer review history of their article (what does this mean?). If published, this will include your full peer review and any attached files.

Reviewer #2: No

---

## [Editor Report · Acceptance letter]

10 Jul 2024

PONE-D-23-38248R1 

PLOS ONE

Dear Dr. Ifeagwu, 

I'm pleased to inform you that your manuscript has been deemed suitable for publication in PLOS ONE. Congratulations! Your manuscript is now being handed over to our production team.

Kind regards, 

on behalf of

Dr. Patrick Christian Ilboudo 

Academic Editor

PLOS ONE